# Optimal Synthesis of Environment-Friendly Iron Red Pigment from Natural Nanostructured Clay Minerals

**DOI:** 10.3390/nano8110925

**Published:** 2018-11-08

**Authors:** Yushen Lu, Wenkai Dong, Wenbo Wang, Junjie Ding, Qin Wang, Aiping Hui, Aiqin Wang

**Affiliations:** 1Key Laboratory of Clay Mineral Applied Research of Gansu Province, Center of Eco-material and Green Chemistry, Lanzhou Institute of Chemical Physics, Chinese Academy of Sciences, Lanzhou 730000, China; luyushen17@mails.ucas.ac.cn or luyushen206@163.com (Y.L.); dongwenkai161@mails.ucas.ac.cn (W.D.); 17693225101@163.com (J.D.); wangqin@licp.cas.cn (Q.W.); aphui1215@163.com (A.H.); 2University of the Chinese Academy of Sciences, Beijing 100049, China; 3Center of Xuyi Palygorskite Applied Technology, Lanzhou Institute of Chemical Physics, Chinese Academy of Sciences, Xuyi 211700, China

**Keywords:** clay mineral, iron red, hybrid pigment, hydrothermal, stability

## Abstract

A series of environment-friendly clay minerals—α-Fe_2_O_3_ iron-red hybrid pigments—were prepared by a simple one-step hydrothermal reaction process using natural nanostructured silicate clay minerals as starting materials. The influence of structure, morphology and composition of different clay minerals on the structure, color properties, and stability of the pigments was studied comparatively by systematic structure characterizations with X-ray diffraction (XRD), Fourier-transform infrared spectroscopy (FTIR), scanning electron microscopy (SEM), transmittance electron microscope (TEM), X-ray fluorescence spectroscopy (XRF), X-ray photoelectron spectroscopy (XPS) and CIE-*L*a*b** Colorimetric analyses. The results showed that the clay minerals act as green precipitants during the hydrothermal reaction to induce in-situ transformation of Fe(III) ions into Fe_2_O_3_ crystals. Meanwhile, they also act as the “micro-reactor” for forming Fe_2_O_3_ crystals and the supporter for inhibiting the aggregation of Fe_2_O_3_ nanoparticles. The color properties of iron-red hybrid pigments are closely related to the surface charges, surface silanol groups, and solid acid sites of clay minerals. The clay minerals with higher surface activity are more suitable to prepare iron-red pigments with better performance. The iron-red hybrid pigment derived from illite (ILL) clay showed the best red color performance with the color values of *L** = 31.8, *a** = 35.2, *b** = 27.1, C* = 44.4 and h° = 37.6, and exhibited excellent stability in different chemical environments such as acid, alkaline, and also in high-temperature conditions.

## 1. Introduction

Pigments make the world in a variety of colors and play an important role in the process of human civilization, which have been used widely in the manufacture of ceramics, sculptures, lacquer wares, and paintings [1,2,3]. In ancient times, the commonly used pigments are mainly natural minerals, such as hematite (α-Fe_2_O_3_), cinnabar (α-HgS), realgar (As_4_S_4_), cadmium sulfide (CdS_1−x_Se_x_), and Egyptian blue (CaCuSiO_10_) [4,5,6]. Hematite is one of the most widely used red mineral pigments because of its abundant resources in nature and changeable color, whose use can be traced back to the Anthropolithic Age [7]. Hematite still plays an important role in the field of pigments, and people are also making great efforts to improve its color, stability, cost, and environmental friendliness. One fascinating idea to improve the properties of mineral pigments is to combine them with substances, e.g., the combination of natural minerals and metal oxides, to produce new hybrid pigments [8]. Based on natural silicate clay minerals, some high-performance pigments have also been synthesized [9]. The incorporation of natural clay minerals into the pigments effectively reduced the production cost, avoided the agglomeration of chromophores, and improved significantly the stability and applicability of pigments [10,11].

Although the basic structural units of different clay minerals are similar, the different combination modes of silica-oxygen tetrahedrons and metal-oxygen octahedrons make them having different morphologies, such as sheets, fibrous, and tubular shape, and the part of Si(IV) in the tetrahedral sheet can be replaced by Al(III), and partial Al(III) and Mg(II) in the octahedral sheet can be replaced by low-valence cations due to isomorphous substitution, resulting in generation of structural or surface negative charges [12]. In addition, the size and shape of pores, specific surface area, crystalline defects, charges, and ion exchange capacities of different clay minerals are also different [13,14], which means that the physical and chemical properties of clay minerals are closely related to their intrinsic microscopic structure, composition, and morphology [15]. In other words, the same types of functional composites prepared from different types of clay minerals may have different structures and properties. For example, in the preparation process of clay mineral-supported nanocomposites such as hybrid materials [10,11,16,17], catalysts [18,19], or composite adsorbents [20,21,22,23,24], the metal ions were firstly deposited on the surface of clay minerals by electrostatic attraction, ion-exchange, chemical complexing or other interactions, and subsequently the precursors were treated by heating or hydrothermal process to generate final products. The formation of precursors and final products is highly dependent on the surface charges and activity of surface groups, so that the types of clay minerals would have great influence on the properties of products. Similarly, in the process of preparing mineral pigments, different clay minerals have different interaction with Fe(III), so that the resultant clay minerals/iron oxides exhibit different morphologies, structure, and color performance. In order to find the optimal clay mineral suitable for the fabrication of high-performance iron-red pigments with vivid color, good stability, low cost, and environmental friendliness, it is of great significance to study systematically the formation process of the hybrid pigment, the interaction of different clay minerals with the iron oxides, and the formation process of such pigments. However, the comparative researches about the effect of different clay minerals on the structure and properties of the resultant iron-red hybrid pigments are rarely concerned.

In order to understand the influence of different clay minerals on the properties of hybrid pigments by a systematic comparative study, in this paper, eight representative natural clay minerals were used as raw materials for synthesis of a series of iron-red hybrid pigments via a one-step hydrothermal process, and the key effect of different clay minerals on the structure and properties of the resultant pigments were studied by Fourier transform infrared spectroscopy (FTIR), X-ray diffraction (XRD), scanning electron microscopy (SEM), transmission electron microscopy (TEM), X-ray fluorescence spectroscopy (XRF), X-ray photoelectron spectroscopy (XPS), and CIE-*L*a*b** Colorimetric analyses. The formation process of the clay minerals-based iron-red hybrid pigments were studied.

## 2. Materials and Methods

### 2.1. Materials

Illite/smectite (IS) clay was taken from Shangsi county of Guangxi Province, China. Rectorite (REC) was provided by Mingliu Rectorite Co. in Wuhan, Hubei province, China. Kaolinite (KAO) was obtained from Longyan City, Fujian Province, China. Montmorillonite (MMT) was provided by Jiashan Baishiwei Biotech Co., Ltd., Jiashan county, Zhejiang Province, China. Vermiculite (VMT) was provided by Lingshou Lihua Mining Processing Factory, Lingshou county, Hebei Province, China. Sepiolite (SEP) was taken from Xiangtan City of Hunan Province, China. Halloysite (HYS) was supplied by Zhengzhou Jinyangguang Ceramics Co., Ltd., Zhengzhou, Henan Province, China. The illite (ILL)-rich clay was taken from Yangtaishan Mine located at Linze county of Gansu Province, China. The chemical composition of the eight clay minerals was listed in Table 1. Iron (III) chloride hexahydrate (FeCl_3_·6H_2_O, AR grade) was purchased from Tianjin Kermel Chemical Reagent Co., Ltd. (Tianjin, China). Deionized water was used to formulate all the aqueous solutions.

### 2.2. Preparation of Iron-Red Hybrid Pigments

The clay mineral powder (4 g) was dispersed to 80 mL of the aqueous solution containing 12 g of FeCl_3_·6H_2_O under continuous stirring at room temperature, followed by a sonication treatment for 10 min to form a homogeneous suspension. The suspension was placed in a 100 mL Teflon-lined stainless steel autoclave, and then the reaction was run at 180 °C for 24 h. After that, the autoclave was naturally cooled to room temperature, and the solid product was separated from the solution by centrifugation at 5000 rpm, thoroughly washed with deionized water, until no Fe(III) ion was detected from the supernatant. Finally, the solid product was dried to constant weight in an oven at 100 °C, ground, and passed through a 200-mesh screen for further use. The as-prepared hybrid pigments were coded as IS-IOR, REC-IOR, KAO-IOR, MMT-IOR, VMT-IOR, SEP-IOR, HYS-IOR, and ILL-IOR, respectively, according to the types of clay minerals (IOR is for iron oxide red).

### 2.3. Measurement of Colorimetric Values

The colorimetric values of the pigments were measured by the colorimetric and reflective spectra, as determined by a Color-Eye Automatic Differential Colorimeter (X-Rite, Ci 7800, Pantone Inc., Carlstadt, NJ, USA). The International Commission on Illumination (CIE) 1976 *L*a*b** colorimetry was used to describe the color of a pigment. In general, *L** is brightness axis (0 is black and 100 is white), the parameters *a** (positive for red, negative for green) and *b** (positive for yellow, negative for blue) indicate the hue or color of the pigment. *C** (chroma) represents the color saturation, calculated by C* = [(*a**)^2^ + (*b**)^2^]^1/2^. h° represents the hue angle, ranging from 0 to 360°, defined as h° = arctan(*b**/*a**).

### 2.4. Stability Tests

The hybrid pigments were fully contacted with 1.0 mol/L of HCl solution, 1.0 mol/L of NaOH solution and anhydrous ethanol by vigorous shaking at 180 rpm for 72 h in a thermostatic shaker (THZ-98A, INESA, Shanghai, China). The pigments were separated from the solution by centrifugation, subsequently washed and dried, and the color values were measured. The chemical stability of the pigments was evaluated by comparing the CIE-*L*a*b** values of the pigments before and after chemical treatment.

A small amount of the pigment powder was dispersed uniformly in an ethanol solution, and then it is uniformly sprayed onto a ceramic substrate. The coated ceramic substrate was calcined at 900 °C, 1000 °C, and 1100 °C, respectively, and the thermal stability of pigment was evaluated by comparing the color values of pigments before and after calcinations.

### 2.5. Characterizations

XRD patterns were collected with an X’Pert PRO diffractometer (PANalytical Co., Almelo, The Netherlands) with a Cu-Kα radiation source (40 kV, 40 mA) in the range of 3° to 80° and at a step interval of about 0.167°. FTIR spectra were collected in the wavenumber region of 4000–400 cm^−1^ with a Nicolet NEXUS FTIR spectrometer (Thermo Electron Corp., Somerset, NJ, USA) using KBr pellets. The surface morphology was observed with a JSM-6701F field emission scanning electron microscope (FESEM, JEOL, Ltd., Tokyo, Japan) in a high vacuum mode at an acceleration voltage of 10.0 kV and a working distance of 10 mm after the sample was fixed on the copper cylinder and then sprayed with gold nanoparticles. TEM was obtained with a JEM-2010 high-resolution transmission electron microscope (TEM, JEOL, Tokyo, Japan) at an acceleration voltage of 200 kV after the sample was ultrasonically dispersed in absolute ethanol and then dropped on a copper mesh. The chemical composition was measured on a MiniPal 4 X-ray fluorescence spectrophotometer (PANalytical Co., Almelo, The Netherlands). The X-ray photoelectron spectroscopy (XPS) was measured with an ESCALab220i-XL electron spectrometer from VG scientific using 300 W Al Ka radiation. TGA curves were determined from 0–800 °C with a Perkin Elmer STA6000 thermogravimetric analyzer (PerkinElmer Co., Ltd., Waltham, MA, USA) at a heating rate of 10 °C/min under a nitrogen atmosphere.

## 3. Results and Discussion

### 3.1. Structure Features of the Red Hybrid Pigments

The crystalline structure of natural clay minerals and the corresponding hybrid pigments (Clay-IORs) were studied by XRD analyses (see Figure 1). The main crystal phase information of natural clay minerals, including IS [25], REC [26], KAO [27], MMT [28], VMT [29,30], SEP [9], HYS [31] and ILL was listed in Appendix A (see Appendix A). As can be seen from Figure 1 and Appendix A, the associated minerals, such as quartz (at 2*θ* = 20.86°, 26.67°, 36.55°, 42.45°, 50.12°, 59.94° and 68.15°) [32], muscovite (at 2*θ* = 8.81°, 17.78°, and 27.96°) [33], talc (at 2*θ* = 9.49°, 18.66° and 28.65°) [30], biotite (at 2*θ* = 10.56°) [29], dolomite (at 2*θ* = 30.94°, 33.54°, 41.13°, 44.95°, and 50.53°), and calcite (at 2*θ* = 23.05°, 29.39°, 35.98°, 39.42°, 47.50°, 48.50°, 56.58°, 57.42°and 60.99°) are present in the natural clay minerals. In REC, MMT, HYS and ILL, relatively lesser associated minerals were observed; but more associated minerals were present in IS, KAO, VMT and SEP. The main associated minerals are quartz for IS, quartz and muscovite for KAO, biotite and talc for VMT, and dolomite and calcite for SEP.

As shown in Figure 1, after the hydrothermal reaction, the characteristic diffraction peaks of REC, KAO, MMT, and ILL weakened, and almost completely disappeared for IS, VMT, SEP, and HYS, indicating that the crystalline framework was damaged due to the removal of partial metal ions during the reaction in acidic condition. The structure integrity of clay mineral is related to the types of center metal ions in tetrahedron or octahedron, and the degree of isomorphism substitution. For example, in acidic solution, the reaction rate of Mg(II) is better than that of Al(III), and Al(III) is better than Si(IV), so the substitution of Mg(II) and/or Fe(III) for Al(III) increases the reaction rate of acid dissolution [34,35]. In addition, non-expanding clay can’t be attacked by protons from the interlayer space, so non-expanding illite and kaolinite are more resistant to acid attack than montmorillonite or vermiculite [36,37]. The characteristic diffraction peaks of the hexagonal hematite crystal phases (JCPDS PDF card 33-0664) can be observed at 2*θ* = 24.21°, 33.19°, 35.72°, 40.96°, 49.55°, 54.18°, 62.54°, and 64.10° in Clay-IORs [38], indicating that Fe(III) were converted in-situ to α-Fe_2_O_3_ in the presence of clay minerals as green precipitants. In addition, the peaks of some associated minerals (i.e., quartz, muscovite, talc) remain in the Clay-IORs with a decreased intensity, suggesting that the associated minerals may also participate in the hydrothermal reaction. 

FTIR spectra of clay minerals and the corresponding pigments were measured to study the properties of hydroxyl groups and identify absorption bands of Si-O-moiety. As shown in Figure 2, the characteristic absorption bands of natural clay minerals can be divided into the following regions: (1) the bands at 3700–3600 cm^−1^ belong to the stretching vibration of structural hydroxyl groups (M-O-H, M′-O-Si-O-H; M, M′ are Al, Mg or Fe); (2) the bands at ~3430 cm^−1^ and ~1630 cm^−1^ are attributed to the stretching vibration and bending vibration of H-O-H of water molecules, respectively; (3) the bands at 1103–1000 cm^−1^ are related to the stretching vibration of Si-O in tetrahedral sheets; (4) the bands at 1100–400 cm^−1^ are concerned with Al-OH, Si-O, and Si-O-Al bonds. The main absorption bands and their specific attributions of the selected natural clay minerals including IS [39], REC [40], KAO [41], MMT [42], VMT [43], SEP [44], HYS [45], and ILL [46], can be referenced in Appendix A (see Appendix A). By comparing the changes in the FTIR spectra before and after the hydrothermal reaction, the structural hydroxyls (Appendix A, M-O-H stretching and M′-O-Si-O-H stretching) weakened and even disappeared after the hydrothermal reaction, indicating that the clay minerals dehydroxylated with the precipitation of octahedral cations. In addition, the Si-O absorption band (Appendix A, Si-O stretching and Si-O-Si stretching vibration) of the tetrahedral sheet weakened and broaden, but it is still present in Clay-IORs, which means that the Si-O tetrahedral sheets were partially destroyed and converted to amorphous Si-O bonds in the Clay-IORs [35]. Meanwhile, the Si-O absorption bands of Si-O_4_ (SiO_2_) (Appendix A) were observed in the FTIR spectra of Clay-IORs, proving that SiO_2_ can be stably present in Clay-IORs. In addition, the characteristic absorption bands of α-Fe_2_O_3_, including the Fe-O-Fe stretching vibration band at about 555 cm^−1^ and the Fe-O-Fe bending vibration band at about 474 cm^−1^, were found in the FTIR spectra of VMT-IOR, SEP-IOR, HYS-IOR and ILL-IOR (Figure 2), which confirmed the presence of α-Fe_2_O_3_ in Clay-IORs [47,48]. The characteristic absorption band of α-Fe_2_O_3_ can’t be found in other pigments due to the low content of α-Fe_2_O_3_.

Figure 3 and Figure 4 showed the SEM and TEM images, respectively. As can be seen, the regular stacking of the structural unit layers of the clay minerals were destroyed, but the primary morphology characteristics were still retained. As demonstrated by XRD analysis, the structural integrity of different clay minerals varies after the hydrothermal reaction. The clay mineral layers in IS-IOR and MMT-IOR are smaller and scattered haphazardly, without regular layer-by-layer stacking. By a comparison, some regularly stacked layers in REC-IOR and VMT-IOR can be found vaguely, and the size of layer is large enough to cover the entire observation region. In contrast, the KAO-IOR and ILL-IOR with a large size and the layer-by-layer stacking extends in one direction, but the layers of ILL are relatively smaller and the stacking is also a little messy. Similarly, fibrous SEP and tubular HYS still remain in SEP-IOR and HYS-IOR, but the dispersion of fibrous SEP is obviously better than that of tubular HYS (Figure 3). What’s more, α-Fe_2_O_3_ can also be observed clearly in all hybrid pigments with a uniform distribution on clay minerals. The α-Fe_2_O_3_ showed a diversity of morphologies, including nanoparticles (IS-IOR, REC-IOR, KAO-IOR, MMT-IOR, VMT-IOR, HYS-IOR and ILL-IOR), nanotubes (REC-IOR, MMT-IOR, and VMT-IOR), and Litchi-like microspheres (VMT-IOR and SEP-IOR) (Figure 3 and Figure 4). In the process of growth of α-Fe_2_O_3_ crystals, nanoparticles were initially formed, and then the nanoparticles aggregated together to form chains of particles, and then assembling to form α-Fe_2_O_3_ nanorods [49], which may be related to the hydroxyl groups on the surface of clay minerals [50]. The further agglomeration of α-Fe_2_O_3_ nanoparticles formed crystals without directionality, which leads to the formation of Litchi-like α-Fe_2_O_3_ microspheres with the minimal surface energy [51]. In addition, the distribution of α-Fe_2_O_3_ on different clay minerals is also different. α-Fe_2_O_3_ nanoparticles and α-Fe_2_O_3_ nanotubes mainly grow on the layers or hollow tube of clay minerals. Litchi-like α-Fe_2_O_3_ microspheres can only be formed in the interstitial spaces of clay minerals due to their large size. Especially, α-Fe_2_O_3_ nanoparticles in ILL-IOR can grow on both the substrate (layers) and the edge face of ILL (Figure 3 and Figure 4). In other words, the distribution of α-Fe_2_O_3_ depends on its size, morphologies and the characteristics of clay minerals.

### 3.2. Chemical Composition of the Hybrid Pigments

The chemical composition of the clay minerals and the corresponding hybrid pigments is shown in Table 1. The clay minerals mainly consist of SiO_2_, Fe_2_O_3_, Al_2_O_3_, MgO, K_2_O, TiO_2_ and CaO, but their specific content are different for different clay minerals. After the formation of hybrid pigments, the contents of SiO_2_, Al_2_O_3_, MgO, K_2_O, TiO_2_, and CaO decreased drastically, while the content of Fe_2_O_3_ increased significantly. The contents of Fe_2_O_3_ are less than 40% for IS-IOR, REC-IOR, KAO-IOR, and MMT-IOR; while the contents of Fe_2_O_3_ are between 57% and 60% for VMT-IOR, HYS-IOR, and ILL-IOR, and SEP-IOR has the maximum Fe_2_O_3_ content of 75.24%. The content of MgO in all hybrid pigments was less than <2%, and even close to zero in KAO-IOR, VMT-IOR, and ILL-IOR. However, the Al_2_O_3_ content in REC-IOR and KAO-IOR are 24.84% and 17.80%, respectively, which is higher than other hybrid pigments. The reason is that the Mg(II) ions are more easily to be removed in acid condition than Al(III) [52,53]. It may also be due to the substitution of Al(III) to Si(IV), and Al(III) in the tetrahedron is more stable than the Al(III) in octahedron in acidic condition [35]. The associated carbonates were dissolved during the hydrothermal reaction in acidic Fe(III) solutions. It was confirmed by XRF results that the main components of the hybrid pigments are silica and Fe_2_O_3_ for IS-IOR, MMT-IOR, VMT-IOR, SEP-IOR, HYS-IOR, and ILL-IOR, and are silica, Fe_2_O_3_ and Al_2_O_3_ for REC-IOR and KAO-IOR.

The XPS full scanning spectra (Appendix A, see Appendix A) and the high-resolution scanning spectra of Si2p, O1s, Al2p and Fe2p of HYS, ILL, HYS-IOR and ILL-IOR (Figure 5) were measured to analyze the change in surface chemical composition. In comparison with HYS and ILL, the Mg and Na elements on the surface of HYS-IOR and ILL-IOR disappeared, and the intensity of the Al element decreased, indicating that the metal elements were leached out. The O and Si elements are stable, and the content of Si elements increases slightly, and the signal of Fe element appears, which proves that HYS-IOR and ILL-IOR are mainly composed of O, Si, Fe elements.

The peaks of Si-O (102.72 eV for HYS, 102.48 eV for ILL, 102.66 eV for HYS-IOR and 102.16 eV for ILL-IOR) and Si-OH (103.29 eV for HYS, 103.09 eV for ILL, 103.31 eV for HYS-IOR and 103.78 eV for ILL-IOR) appeared in the Si2p spectra, and the Al2p showed two types of peaks, Al-O (74.24 eV for HYS, 74.61 eV for ILL, 74.26 eV for HYS-IOR and 74.15 eV for ILL-IOR) and Al-OH (74.86 eV for HYS, 74.84 eV for HYS-IOR and 74.85 eV for ILL-IOR). However, ILL differs from HYS in that Al2p spectra of ILL only showed Al-O peaks. After the hydrothermal reaction, the peak area of Si-OH increased while peak area of Si-O decreased after the conversion of ILL to ILL-IOR; but the change of peak area is not obvious after the conversion of HYS to HYS-IOR. The change of Al-O to Al-OH is more pronounced after the conversion of ILL to ILL-IOR. It was illustrated that the hydrothermal reaction is accompanied with a hydroxylation process on the surface of clay mineral. The significant Fe_2_O_3_ peaks were observed in the Fe2p spectra of HYS-IOR (724.60 eV for 2p1/2 and 711.23 eV for 2p3/2) and ILL-IOR (724.70 eV for 2p1/2 and 711.33 eV for 2p3/2), which further confirmed the formation of Fe_2_O_3_. The Fe2p binding energy of Fe_2_O_3_ in ILL is larger than that of ILL-IOR, indicating that the Fe element in ILL is mainly in form of structural iron (725.60 eV for 2p1/2 and 712.31 eV for 2p3/2) [54], while the Fe_2_O_3_ in HYS-IOR and ILL-IOR mainly grows on the surface of clay minerals. The O1s spectra of HYS and ILL were obtained by fitting SiO_2_ (532.43 eV for HYS and 532.26 eV for ILL) and Al_2_O_3_ (532.09 eV for HYS and 531.65 eV for ILL), but new peaks Fe/SiO_2_ (532.93 eV for HYS-IOR and 533.16 eV for ILL-IOR) appeared in addition to SiO_2_ (532.75 eV for HYS-IOR and 532.61 eV for ILL-IOR) and Al_2_O_3_ (531.98 eV for HYS-IOR and 531.15 eV for ILL-IOR) in HYS-IOR and ILL-IOR. At the same time, Si-O shifts to Fe/SiO_2_ and Al-O is away from Fe/SiO_2_, indicating that the bonding mode of Si-O and Al-O bonds changed. The appearance of Fe/SiO_2_ and the chemical shifts of Si-O and Al-O in different directions indicate that the formed Fe_2_O_3_ crystals are fixed on the surface of clay minerals by Fe-O-Si or Fe-O-Al bonds. Moreover, the region where Fe/SiO_2_ overlaps with Si-O is larger than the region overlapping with Al-O, indicating that Fe-O-Si can be formed more easily. This also proves that Fe_2_O_3_ grows in-situ on the surface of clay minerals through the Si-O-Fe bond.

### 3.3. Color Properties of the Hybrid Pigments

Figure 6 showed the digital photograph of the used clay minerals and corresponding hybrid pigments. As can be seen, the clay mineral powder presented different colors: light goldenrod yellow for IS, black for REC, white for KAO and SEP, palegoldenrod for MMT, peru for VMT, moccasin for HYS and khaki for ILL, respectively. The difference in color is due to the different chemical composition, and ingredients of associated minerals in the clay minerals, as shown by XRD (Figure 1 and Appendix A, see Appendix A) and XRF (Table 1) analyses. Natural clay minerals react with Fe(III) under the hydrothermal condition to produce hybrid pigments with various colors. IS-IOR, REC-IOR, KAO-IOR, MMT-IOR, VMT-IOR, SEP-IOR, HYS-IOR and ILL-IOR samples showed rosy brown, brown, Indian red, sandy brown, tomato, firebrick, dark red, and red colors, respectively.

The color-values of the hybrid pigments were determined by CIE-*L*a*b** colorimetric method to evaluate the color performance. As shown in Table 2, *L** values of the hybrid pigments are 43.5 for IS-IOR, 40.9 for REC-IOR, 42.9 for KAO-IOR, 44.5 for MMT-IOR, 39.0 for VMT-IOR, 28.4 for SEP-IOR, 27.5 for HSY-IOR, and 31.8 for ILL-IOR, indicating a slight difference in brightness. However, *a** values of different Clay-IORs are quite different, with the minimum value of 10.4 for IS-IOR and the maximum value of 35.2 for ILL-IOR. The *a** values of other pigments are 10.9 for REC-IOR, 12.1 for KAO-IOR, 15.4 for MMT-IOR, 20.2 for VMT-IOR, 22.0 for SEP-IOR and 29.6 for HYS-IOR. Similarly, *b** values are in the range from 3.8 (for IS-IOR) to 27.1 (for ILL-IOR). The comprehensive color-values (Table 2) and appearance colors (Figure 6) of the hybrid pigments were compared with commercial iron-red pigments (COM-IOR), it was found that HSY-IOR and ILL-IOR have better color performance. Among them, ILL-IOR (*L** = 31.8, *a** = 35.2, *b** = 27.1, C* = 44.4 and h° = 37.6) has the best *a** value of 35.2, the highest C*** value of 44.4 and the moderate *L** value of 31.8, showing a pure red color. These results suggest that the performance of iron-red hybrid pigments is highly dependent on the structure, morphology and components of clay minerals.

Colors of iron-red hybrid pigments can also be described easily and accurately by labeling color coordinates (x, y) in the CIE1931 chromaticity diagram. As shown in Figure 7, the coordinate position of points a, b, c, and d are far from the red area, indicating that the red tone is not significant in IS-IOR, REC-IOR, KAO-IOR, and MMT-IOR. *a**-value of KAO-IOR is larger than that of REC-IOR (12.9 > 10.9), but the position of point-c is more far from the red region than the point-b, which is caused by the small color saturation of KAO-IOR over REC-IOR (12.6 < 15.8). The point e and f at the edge of the red area, suggesting that the colors of VMT-IOR and SEP-IOR begin to appear red. Points g and h fall in the red area, reflecting the red tone dominated in HYS-IOR and ILL-IOR. The results are consistent with the visual color of the pigment and the color parameters determined by the CIE-*L*a*b** colorimetric method.

The colors of the hybrid pigments can also be characterized by UV-vis diffuse reflectance spectroscopy in the visible wavelength range. As shown in Appendix A, in the wavelength range of 360–550 nm, the reflectance of IS-IOR, REC-IOR, KAO-IOR, MMT-IOR, and VMT-IOR is between 5% and 13%, while the reflectance of SEP-IOR, HYS-IOR, and ILL-IOR is extremely low, about 2%. In the range of 550–575 nm, there are absorption peaks with different intensity, indicating that the hybrid pigments mainly absorb green light, while complementary color of red is green. In the wavelength region of 575–750 nm, the reflectivity sharply rises, and the intensity is ranged from 21% to 42%. All these indicate that the hybrid pigments exhibit different degrees of red color. Defining *N = R*_740_*/R*_450_, 740 nm for red light, 450 nm for blue light, to reflect the reflectance difference in 360–550 nm and 575–750 nm. By calculation, the *N*-value is in the order: ILL-IOR (19.35) > HYS-IOR (11.32) > SEP-IOR (7.71) > VMT-IOR (5.98) > MMT-IOR (4.80) > KAO-IOR (2.98) > IS-IOR (2.82) > REC-IOR (2.64). The order of *N* values is almost the same as the order of *a** values (Table 2). When *N*-value is small, the pigment can’t reflect the light of a specific wavelength, forming a complex color, which also explains the difference in the visual color of the hybrid pigments (Figure 6).

### 3.4. Stability of the Hybrid Pigments

#### 3.4.1. Chemical Stability

The hybrid pigments and commercial iron-red pigment were immersed in an absolute ethanol solution, 1 mol/L HCl solution and 1 mol/L NaOH solution for 72 h, and the change of *a**-value before and after the treatment was compared to evaluate the chemical stability. As shown in Figure 8, after treatment with absolute ethanol, *a**-value of all the pigments showed no obvious change, indicating that the hybrid pigments are stable in the organic solvents. After treatment with NaOH solution, *a**-value decreased, and this tendency is more pronounced after treatment with HCl solution. Especially, *a**-HCl value is only 0.8 for REC-IOR. Compared with other pigments, REC-IOR has the highest Al_2_O_3_ content of 24.84% (Table 1), which could be dissolved out in the acid or alkali solution, causing a change of structure and color. Surprisingly, ILL-IOR has excellent color performance in a variety of chemical environments, with *a** = 35.2, *a**-A.E. = 31.1, *a**-NaOH = 28.6 and *a**-HCl = 26.7. All *a** values are higher than the initial *a**-value of COM-IOR (*a** = 26.9). The main components of ILL-IOR are Fe_2_O_3_ and SiO_2_, so that the silica composition in the hybrid pigments is very important for its chemical stability. The silica skeleton of the clay mineral is converted into a silica supporter in the process of synthesizing the hybrid pigment, which is beneficial to improve the stability of the pigments.

#### 3.4.2. Thermal Stability

As shown in Figure 9, the hybrid pigments formed a stable and uniform coating on ceramic substrate. After calcination at high temperatures, the shape is intact and no peeling occurs. After calcinations at 900 °C, the appearance color of the coating changed significantly. The coating colors of IS-IOR and REC-IOR have undergone a fundamental change. For IS-IOR, from rosy brown to saddle brown, the red tone increases. For REC-IOR, the yellow tone increases, from brown to sandy brown. This may be due to the change in the crystal form of some components of the hybrid pigments during the high-temperature treatment, especially for IS-IOR and REC-IOR with a high content of Al_2_O_3_. After calcinations at 1000 °C, the colors of coating further deepened. After calcinations at 1100 °C, the colors become darker and the brightness decreased. Above the glass transition temperature of SiO_2_ (Tg~1000 °C), the SiO_2_ skeleton becomes soft, and the Fe_2_O_3_ particles are aggregated together, resulting in deterioration of properties [55], which caused the above phenomenon. At the same time, the change in *a**-value is consistent with the color change of the coatings (Figure 8). After calcination treatment at 900 °C, the color performance was improved, and the heating is continued at a higher temperature, and the color performances lowered. However, *a*-*value is still higher than that of the corresponding pigments. That is to say, the hybrid pigments have good thermal stability. In addition, the TGA results (Appendix A, see Appendix A) of ILL and ILL-IOR also demonstrated that the iron red pigments prepared from natural nanostructured clay minerals such as ILL showed better thermal stability than raw clay minerals because the less weight loss of the pigments.

### 3.5. Proposed Formation Process of Hybrid Pigments

The formation process of clay minerals/α-Fe_2_O_3_ iron-red hybrid pigments was proposed according to the XPS analyses of HYS and ILL examples and shown as follows.
(1a)>Si+OH=SiOH
(1b)>SiO+H=SiOH
(1c)> Al +OH=AlOH
(1d)>AlO+H=AlOH
(2)$$a\;(>\mathrm{SOH})+p\;{\mathrm{Fe}}^{3+}+y\;{\mathrm{OH}}^{-}={\left(>\mathrm{SO}\right)}_{a}{\mathrm{Fe}}_{3}{{\left(\mathrm{OH}\right)}_{y}}^{(3z-a-y)+}$$
(3)$${\left(>SO\right)}_{a}{\mathrm{Fe}}_{3}{{\left(\mathrm{OH}\right)}_{y}}^{(3z-a-y)+}+\left(3z-a-y\right)H^{-}={\mathrm{SiO}}_{2}/{\mathrm{Fe}}_{2}O_{3}\;(\text{Si-O-Fe})$$

The mineral bulk phase lattice is truncated at the surface to form a dangling bond, and the dangling bond itself or the structural recombination forms a surface group [56]. As shown in XPS spectra, there are Al-O, Si-O, Al-OH, and Si-OH on the surface of clay minerals [57]. Once the clay minerals are in contact with water, a hydroxylation reaction occurs on the surface to form a hydroxyl functional group (Equations (1a)–(1d)) [58]. The surface hydroxyl groups, AlOH and SiOH (abbreviated as SOH, S=Si or Al). The SiOH reacts with the Fe(III) in solution by electrostatic interaction, as shown in Equation (2) [58,59]. In this process, SiOH acts as a Bronsted acid, it is negatively charged after giving protons, and generates electrostatic interaction with Fe(III) in the solution. The silanol group intensity is larger than alcohol group, which is easier to give a proton than the aluminum alcohol group, so the surface coordination reaction is dominated by the silanol group to form more (>SiO)_a_ Fe_3_ (OH)_y_^(3z−a−y)+^. The surface hydroxyl group is an amphoteric functional group, and a hydroxyl group which does not undergo surface coordination reaction can accept protons and further induce the formation of Fe_2_O_3_ (Si-O-Fe) (Equation (3)) [44,46]. The formation process of hybrid pigments is accompanied with the dissolution of octahedral cations and the transformation of clay minerals into the silica.

### 3.6. Effects of Different Clay Minerals on Iron Red Hybrid Pigments

The above analysis implies that the surface of clay mineral is the “micro-reactor” for the growth of Fe_2_O_3_ crystal, which is a key to the formation of hybrid pigments. There are functional hydroxyl groups, surface charges, solid acid sites on the surface of clay minerals [56,58]. The structure, composition, and morphology of the clay minerals and the medium environment co-determine the surface reactivity. In this study, the clay minerals are in the same aqueous solution of FeCl_3_, so the medium environment is assumed to be the same. This paper mainly explores the effects of different clay minerals on iron red hybrid pigments.

Appendix A lists the structural information and *a**-values of raw clay minerals [60] and corresponding hybrid pigments. As can be seen, 2:1 type clay minerals are better than 1:1 type and mixed-layer clay minerals for preparation of clay-based iron-red hybrid pigments. Moreover, *a** values of the hybrid pigments are linearly related to net layer charge per formula unit (ξ). In the 1:1 type KAO, there is little substitution in the crystal lattice, and the permanent charge is very less. The active sites on KAO are only Si-OH, Al-OH, and Lewis acid sites are at the edge of the layer, but the edge area is very small [61]. The 2:1 type clay minerals have a large amount of cationic isomorphous substitutions that produce more permanent charge [12,61]. And there are more Si-O, Si-OH groups on the surface. That is, the surface reactivity of 2:1 type is higher than 1:1 type clay minerals. In addition, the *a** values of SEP-IOR and HYS-IOR are higher, because the fiber rod and the hollow tube have both an outer surface and an inner surface, and have a large specific surface area, providing a rich reactive site.

The effect of structure of clay mineral on iron red hybrid pigments was also reflected in the regulation of α-Fe_2_O_3_ morphology. The hydrothermal process may activate clay minerals, increase the specific surface area, porosity, and surface acidity. The more hydroxyl groups on the surface of clay minerals, the more nucleation sites are used for formation of α-Fe_2_O_3_. Because the crystal growth space is limited, α-Fe_2_O_3_ nanoparticles are easily formed. On the contrary, when the surface hydroxyl group is less, the crystal growth space is sufficient, which make the α-Fe_2_O_3_ nanoparticles further grow to form nanorods, and Litchi-like microspheres. The aggregation of α-Fe_2_O_3_ leads to the decrease of properties of pigment.

The composition of the clay minerals also affects the surface components, especially the nature of the surface atoms, further affecting the performance of hybrid pigments [62]. The main bonding mode of clay-based hybrid pigments, Al-O-Fe or Si-O-Fe, can be judged from the composition of hybrid pigments (Table 1). When hybrid pigment contains a large amount of Al_2_O_3_ in addition to SiO_2_ and Fe_2_O_3_, the main bonding modes are Al-O-Fe or Si-O-Fe. When the content of Al_2_O_3_ is less, Si-O-Fe dominates the properties of hybrid pigments. It has been proved by XPS analysis that the effect of Al-OH is smaller than that of Si-OH during the in-situ growth process of Fe_2_O_3_ on the surface of clay minerals. What’s more, the analysis of the stability implies that Al-O-Fe bonds are not stable. Therefore, the hybrid pigments mainly bonded by Si-O-Fe have better performance. In addition, Fe_2_O_3_ is a chromogenic substance, and its effect on the color of the pigment is also obvious. There are a handful of Fe_2_O_3_, the tone of the pigment is less reddish, and as the increase in content of Fe_2_O_3_, the red color increases; while a further increase will lead to the aggregation of Fe_2_O_3_ and the reduction of brightness, showing a dark red color.

In summary, illite (ILL) is a typical 2:1 type clay mineral, with Al(III) substituted for Si(IV) in the crystal lattice, resulting in more negative charge and higher surface reactivity. Since the isomorphism occurs in the tetrahedral layer, the negative charges generated are close to the surface of the crystal layer, so the layer generates a strong electrostatic attraction, making the structure more stable, which is not easily eroded by acid [36,63]. After the hydrothermal reaction, the regular stack of tetrahedral and octahedral layers in ILL is destroyed, but the structural integrity is still high, and regular sheets structures exist, making the growth space of α-Fe_2_O_3_ crystals is limited and can only grow to be a nanoparticle. The iron red hybrid pigment with uniform loading of α-Fe_2_O_3_ nanoparticles on ILL layer by Si-O-Fe bonding and main components of Fe_2_O_3_ and SiO_2_ was synthesized from ILL, which has excellent color performance (*L** = 31.8, *a** = 35.2, *b** = 27.1, C* = 44.4 and h° = 37.6), showing a pure red color and excellent stability in different chemical environments.

## 4. Conclusions

Natural clay minerals are suitable for preparing iron-red hybrid pigments because they possess surface silanol groups, surface charges, solid acid sites, and also huge reserves in nature. However, the surface reactivity of different clay minerals is different due to the differences in structure, morphology, and composition, which have great influence on the properties of pigments. We studied systematically the action of eight representative natural clay minerals in fabricating iron-red pigments, and achieved the optimal synthesis of the hybrid pigment by selecting the optimal clay mineral. It has been confirmed the higher surface reactivity, determined by the structure, composition and morphology of the minerals, is favorable to the formation of iron-red hybrid pigments. The formation process of the iron-red hybrid pigment is: (1) hydroxylation of clay mineral to form a surface hydroxyl group; (2) the hydroxyl group undergoes a surface coordination site to react with Fe(III); (3) H^+^ induces the formation of iron red hybrid pigments. During this period, parts of the cations in octahedral sheet are dissolved out, and the tetrahedral sheet converted to a siliceous supporter to control the growth of Fe_2_O_3_. The iron-red hybrid pigment prepared using illite (ILL) showed a pure red color and the best properties, with *L** = 31.8, *a** = 35.2, *b** = 27.1, C* = 44.4 and h° = 37.6. In addition, the pigment is stable in different chemical environments and even high temperature condition. The comparative study of different clay minerals for synthesis of iron-red hybrid pigments laid a solid foundation for preparation of cost-efficient and vivid mineral pigments, and would also effectively promote the high value utilization of clay mineral resources.

## Figures and Tables

**Figure 1 nanomaterials-08-00925-f001:**
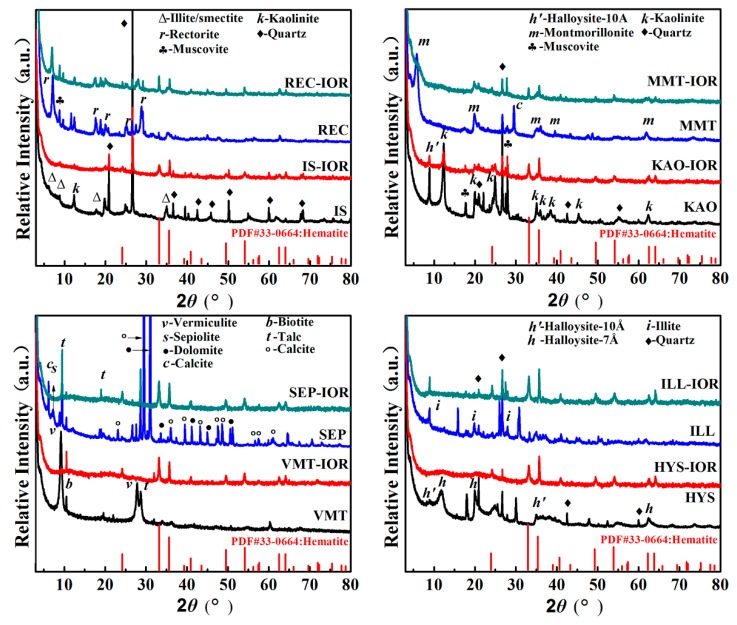
XRD patterns of selected natural clay minerals and the corresponding hybrid pigments.

**Figure 2 nanomaterials-08-00925-f002:**
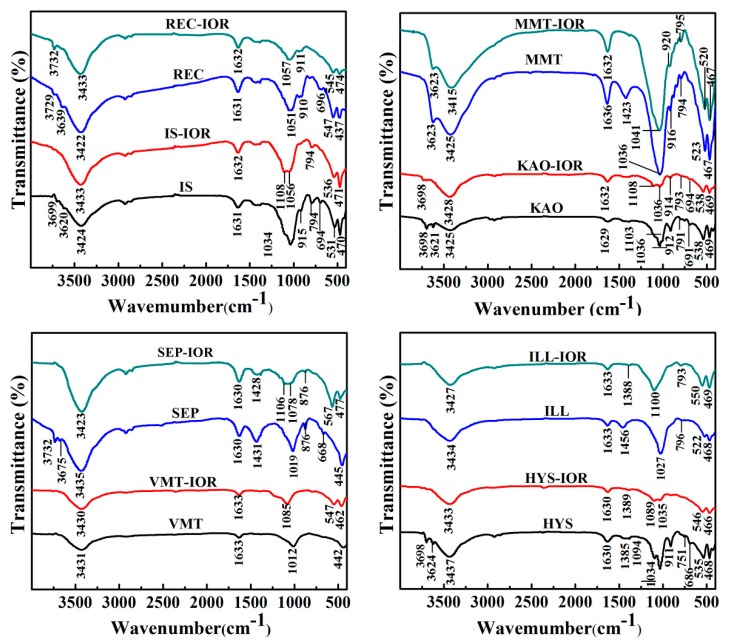
FTIR spectra of selected the natural clay minerals and the corresponding hybrid pigments.

**Figure 3 nanomaterials-08-00925-f003:**
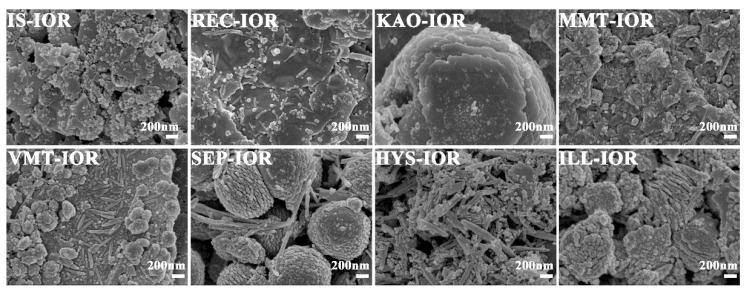
SEM images of IS-IOR, REC-IOR, KAO-IOR, MMT-IOR, VMT-IOR, SEP-IOR, HYS-IOR, and ILL-IOR.

**Figure 4 nanomaterials-08-00925-f004:**
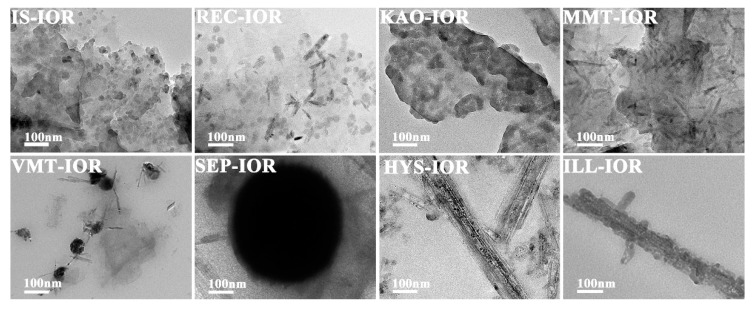
TEM images of IS-IOR, REC-IOR, KAO-IOR, MMT-IOR, VMT-IOR, SEP-IOR, HYS-IOR, and ILL-IOR.

**Figure 5 nanomaterials-08-00925-f005:**
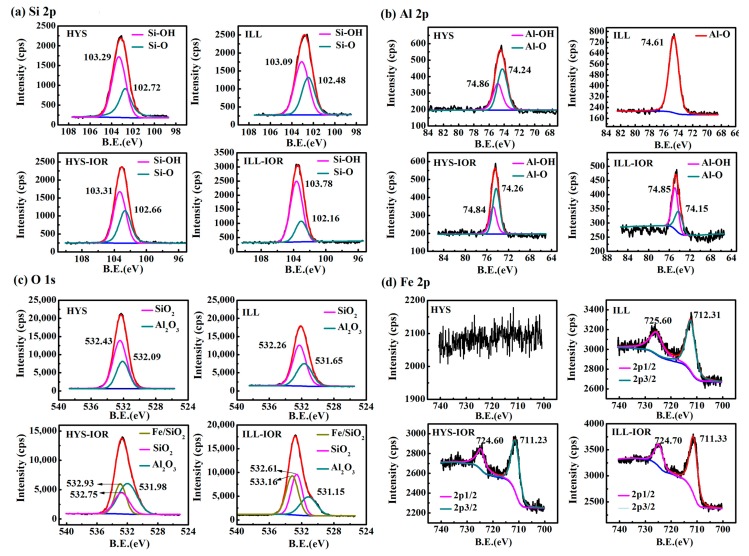
X-ray photoelectron spectroscopy (XPS) high-resolution scanning spectra of Halloysite (HYS), illite (ILL), HYS-IOR and ILL-IOR: (**a**) Si2p, (**b**) Al2p, (**c**) O1s, and (**d**) Fe2p.

**Figure 6 nanomaterials-08-00925-f006:**
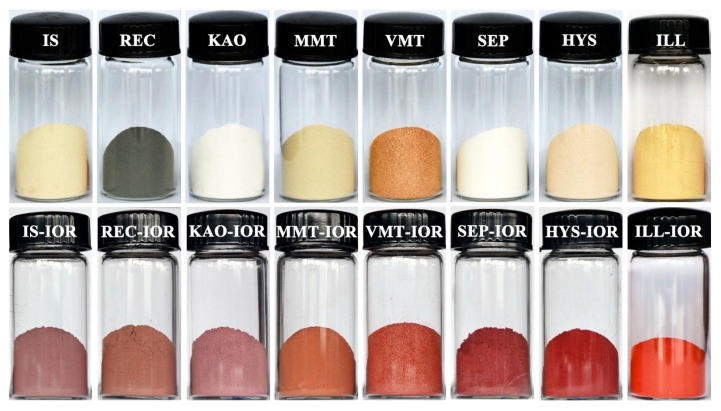
Digital photos of the used natural clay minerals and the corresponding hybrid pigments (For interpretation of the references to color in this figure legend, the reader is referred to the web version of this article).

**Figure 7 nanomaterials-08-00925-f007:**
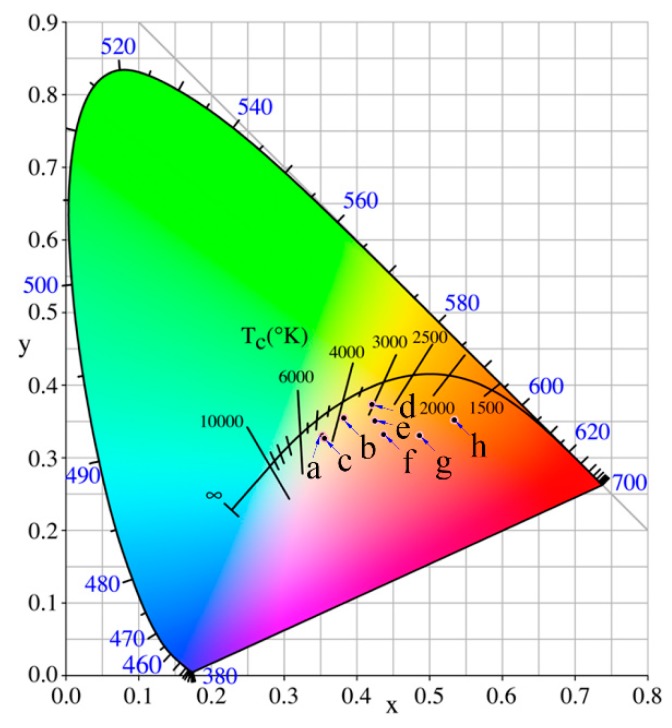
Chromatic CIE coordinates of iron-red hybrid pigments (a: IS-IOR, b: REC-IOR, c: KAO-IOR, d: MMT-IOR, e: VMT-IOR, f: SEP-IOR, g: HYS-IOR, and h: ILL-IOR).

**Figure 8 nanomaterials-08-00925-f008:**
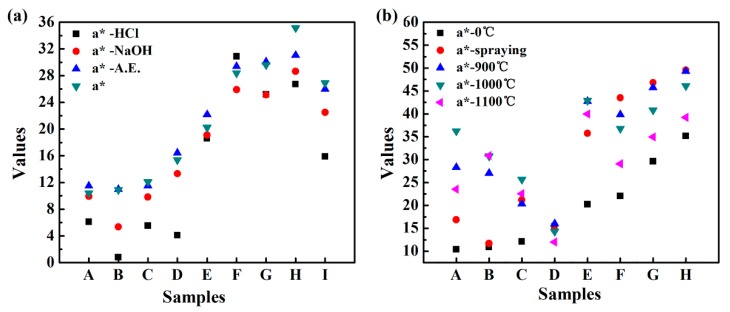
*a**-value of iron red hybrid pigments after treatment under different conditions: (**a**) different chemical environments and (**b**) calcined coating at high temperature (A: IS-IOR, B: REC-IOR, C: KAO-IOR, D: MMT-IOR, E: VMT-IOR, F: SEP-IOR, G: HYS-IOR, H: ILL-IOR, I: COM-IOR).

**Figure 9 nanomaterials-08-00925-f009:**
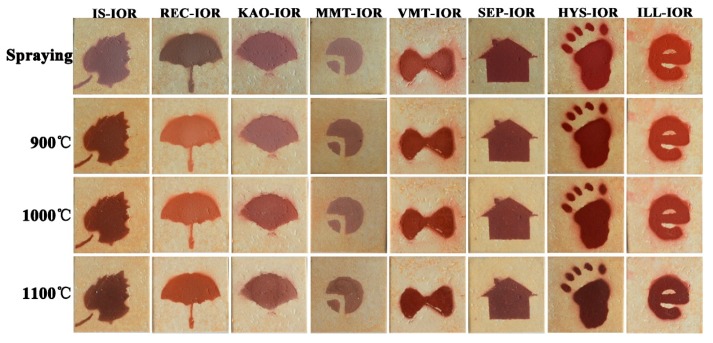
Digital photos of sprayed coatings on ceramic substrate after calcinations at different temperatures (900 °C, 1000 °C, and 1100 °C).

**Table 1 nanomaterials-08-00925-t001:** Chemical compositions of raw clay minerals and the corresponding hybrid pigments (mass%). The as-prepared hybrid pigments were coded as IS-IOR, REC-IOR, KAO-IOR, MMT-IOR, VMT-IOR, SEP-IOR, HYS-IOR, and ILL-IOR, respectively, according to the types of clay minerals (IOR is for iron oxide red).

Samples	SiO_2_	Fe_2_O_3_	Al_2_O_3_	MgO	K_2_O	TiO_2_	CaO
IS	64.58	5.18	22.95	1.25	4.45	1.32	0
IS-IOR	46.52	39.16	9.71	0.35	1.92	0.84	0
REC	40.00	4.66	31.85	0.07	1.42	4.68	9.90
REC-IOR	37.39	25.83	24.84	0.12	1.10	3.84	3.84
KAO	57.57	0.26	37.84	0.00	3.82	0.03	0
KAO-IOR	41.93	36.80	17.80	0.03	2.34	0	0
MMT	51.42	8.04	14.15	2.91	0.25	0.88	21.04
MMT-IOR	48.05	37.19	9.74	1.74	0.17	0.65	0.55
VMT	40.50	19.35	15.18	11.18	2.21	2.47	8.24
VMT-IOR	34.66	59.06	1.98	0.22	0.09	1.62	0.71
SEP	20.45	0.60	1.25	11.97	0.21	0.07	64.13
SEP-IOR	20.23	75.24	1.03	1.98	0.05	0	0.69
HSY	48.79	1.82	40.49	0.00	1.18	0	1.77
HSY-IOR	29.38	59.24	7.95	0.00	0.38	0	0
ILL	48.17	11.11	16.30	4.23	4.01	1.32	10.85
ILL-IOR	34.21	57.88	4.45	0.11	1.25	0.84	0

**Table 2 nanomaterials-08-00925-t002:** CIE-*L**, *a**, *b**, C* and h° values of the iron-red hybrid pigments.

Samples	*L**	*a**	*b**	C*	h°
IS-IOR	43.5	10.4	3.8	11.1	20.3
REC-IOR	40.9	10.9	11.4	15.8	46.3
KAO-IOR	42.9	12.1	3.4	12.6	15.5
MMT-IOR	44.5	15.4	20.8	25.9	53.5
VMT-IOR	39.0	20.2	15.2	25.3	37.0
SEP-IOR	28.4	22.0	10.3	24.3	25.1
HSY-IOR	27.5	29.6	15.1	33.2	27.0
ILL-IOR	31.8	35.2	27.1	44.4	37.6
COM-IOR	35.5	26.7	19.2	32.9	35.7

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
