# Peer review of "Optimal Synthesis of Environment-Friendly Iron Red Pigment from Natural Nanostructured Clay Minerals"

_nanomaterials, 2018, doi:10.3390/nano8110925_

Round 1
Reviewer 1 Report
Dear Editor,
the paper by Yushen et al. entitled “Optimal Synthesis of Environment-friendly Iron Red Pigment from Natural Nanostructured Clay Minerals” comprehensively elucidates how eight clay minerals act as precipitant of Fe(III) ions as Fe2O3 crystals for the preparation of iron-red hybrid pigments. Also interesting, how they hinder the aggregation of Fe2O3 particles. The subject fits to the journal scope and, in my opinion, is very interesting for the scientific community dealing with clay-composites like me. This paper is well done.
The introduction is well written and provided by proper references.
The discussion is for sure exhaustive.
Conclusions are clear.
From the scientific point of view I have no objections. Afterwards, there only several forma issues.
Minor requests:
1) In the SEM images all the scale bars and all the other information are not readable…. Please solve the problem.
2) From line 234…. This description is completely useless, all the info are table 1. Please choose, or the table or the description in the text.
3) Thermal stability part: TGA (Thermo gravimetric analysis) of the samples could be very interesting. (of course, just if it is possible)
Based on the above arguments, I recommend the paper for publication after minor revision.
Author Response
Dear Prof. Pan and Reviewer,
We are very grateful for your sincere suggestions that greatly contribute to improve our manuscript entitled “Optimal Synthesis of Environment-friendly Iron Red Pigment from Natural Nanostructured Clay Minerals” (Manuscript ID: nanomaterials-382764). We have carefully revised the manuscript according to your suggestions. The details about the revisions are shown in the revised manuscript, and the answers to the reviewer’s questions are shown as follows one by one. All changes made to the manuscript are marked as “highlight” for easy check/editing purpose.
Responses to Reviewer #1
Question (1): In the SEM images all the scale bars and all the other information are not readable…. Please solve the problem.
Author’s reply: Thank you for your suggestion. We have redrawn the scale bars in the SEM images, and the newly revised SEM images were shown in Figure 3 in the revised manuscript.
Question (2): From line 234…. This description is completely useless, all the info are table 1. Please choose, or the table or the description in the text.
Author’s reply: Thank you for your suggestion. We have removed the text that is duplicated with the data listed in Table 1 according to your suggestion.
Question (3): Thermal stability part: TGA (Thermo gravimetric analysis) of the samples could be very interesting. (of course, just if it is possible)
Author’s reply: Thanks for your suggestion. We have analyzed the TGA curves of illite (ILL) and the optimal hybrid pigment (ILL-IOR) (Fig. R1). As can be seen, the weight loss process of ILL is divided into two parts: (i) the weight loss about 6% below 400 °C is due to lose of the adsorbed water and the interlayer water; (ii) The weight loss of about 9% at the temperature above 400 °C is ascribed to the dehydration of the hydroxyl group of illite. What’s more, weight loses of ILL-IOR about 6% during the whole process, which is close to the first stage of ILL, and there is no obvious weight loss step. It indicates that the mass loss in ILL-IOR may be due to the loss of adsorbed water and interlayer water in illite. ILL-IOR has less weight loss, which proved that ILL-IOR has better thermal stability than ILL. According to your suggestion, we supplemented the corresponding discussion in the revised manuscript, and added the TGA figure in the supplementary material.
Fig. R1. TGA curves of ILL and ILL-IOR.
In addition, the context was checked carefully and the improper phrase or sentences were corrected.
Finally, thanks for your intensive guidance to our manuscript. If you have any question about the revised manuscript, please do not hesitate to contact me.
Best Wishes
Sincerely
Prof. Aiqin Wang,
Lanzhou Institute of Chemical Physics,
Chinese Academy of Sciences,
Lanzhou 730000,
P. R. China,
Tel: 86 931 4968118
Fax: 86 931 4968019
E-mail: aqwang@licp.cas.cn

Reviewer 2 Report
The manuscript deals with clay/dye composites and their structure/stability investigations. The topic is interesting and there are new aspects worth of being published. Introduction is not up-to-date.
-Relevant recent references on clay based hybrid materials are missed (Nanomaterials. 7 (2017) 199–210. doi:10.3390/nano7080199; Langmuir. 33 (2017) 3317–3323. doi:10.1021/acs.langmuir.7b00600) as well as some more specific papers on dye adsortpion (Appl. Sci. 2018, 8(4), 608; J. Phys. Chem. C 2016, 120 (25), 13492–13502; Appl. Clay Sci. 2013, 72 (0), 132–137.).
- SEM/TEM experimental details should be added.
- Table 1 reports iron content that are duplicated in pag 8 within the text. Please revise.
- The paragraph “Formation mechanism of hybrid pigments” is hard to follow based on the results.
- Fig 4 should be TEM and not SEM images as stated in the caption.
Author Response
Dear Prof. Pan and Reviewer,
We are very grateful for your sincere suggestions that greatly contribute to improve our manuscript entitled “Optimal Synthesis of Environment-friendly Iron Red Pigment from Natural Nanostructured Clay Minerals” (Manuscript ID: nanomaterials-382764). We have carefully revised the manuscript according to your suggestions. The details about the revisions are shown in the revised manuscript, and the answers to the reviewer’s questions are shown as follows one by one. All changes made to the manuscript are marked as “highlight” for easy check/editing purpose.
Responses to Reviewer #2
Question (1): Relevant recent references on clay based hybrid materials are missed (Nanomaterials. 7 (2017) 199–210. doi:10.3390/nano7080199; Langmuir. 33 (2017) 3317–3323. doi:10.1021/acs.langmuir.7b00600) as well as some more specific papers on dye adsortpion (Appl. Sci. 2018, 8(4), 608; J. Phys. Chem. C 2016, 120 (25), 13492–13502; Appl. Clay Sci. 2013, 72 (0), 132–137.).
Author’s reply: Thanks for your suggestion! We agree with your suggestion that these references are necessary and helpful to explain or support our experiment results. According to your suggestion, we have cited these references and updated the reference lists in the revised manuscript.
Question (2): SEM/TEM experimental details should be added.
Author’s reply: Thanks for your suggestion! We added SEM and TEM experiment details in the revised manuscript.
Question (3): Table 1 reports iron content that are duplicated in pag 8 within the text. Please revise.
Author’s reply: Thanks for your suggestion! We have modified the corresponding improper description according to your suggestion in the revised manuscript.
Question (4): The paragraph “Formation mechanism of hybrid pigments” is hard to follow based on the results.
Author’s reply: Thanks for your suggestion. We changed the paragraph “Formation mechanism of hybrid pigments” to “Proposed Formation process of hybrid pigments“, and modified the relevant discussion according to your suggestion, it may be more appropriate. Thank you again!
Question (5): Fig 4 should be TEM and not SEM images as stated in the caption.
Author’s reply: Thanks for your reminder! We corrected this error according to your sincere suggestion.
In addition, the context was checked carefully and the improper phrase or sentences were corrected.
Finally, thanks for your intensive guidance to our manuscript. If you have any question about the revised manuscript, please do not hesitate to contact me.
Best Wishes
Sincerely
Prof. Aiqin Wang,
Lanzhou Institute of Chemical Physics,
Chinese Academy of Sciences,
Lanzhou 730000,
P. R. China,
Tel: 86 931 4968118
Fax: 86 931 4968019
E-mail: aqwang@licp.cas.cn

Reviewer 3 Report
This paper on the synthesis of iron red pigments from clays is well structured, written and the discussion consistent. Minor issues should be improved:
1. A main conclusion is that iron-red from illite clay shows the best red color. It is similar to the case of the color of “terra sigillata” pottery form the roman culture. As shown in reference DOI: 10.1039/c4ja00367e (Evolution of terra sigillata technology from Italy toGaul through a multi-technique approach) the red color of this “terra sigillata” comes from the decomposition of the illite clay after firing and the formation of hematite with high degree of substitution by Al and Ti and high crystallinity. Since the presence of hematite is confirmed (Figure 1) Titanium analysis by ICP would have be interesting to perform on the samples.
2. Please try not to reproduce the content of the tables into the text of the discussion (lines 233-242). It is too repetitive.
Author Response
Dear Prof. Pan and Reviewer,
We are very grateful for your sincere suggestions that greatly contribute to improve our manuscript entitled “Optimal Synthesis of Environment-friendly Iron Red Pigment from Natural Nanostructured Clay Minerals” (Manuscript ID: nanomaterials-382764). We have carefully revised the manuscript according to your suggestions. The details about the revisions are shown in the revised manuscript, and the answers to the reviewer’s questions are shown as follows one by one. All changes made to the manuscript are marked as “highlight” for easy check/editing purpose.
Responses to Reviewer #3
Question (1): A main conclusion is that iron-red from illite clay shows the best red color. It is similar to the case of the color of “terra sigillata” pottery form the roman culture. As shown in reference DOI: 10.1039/c4ja00367e (Evolution of terra sigillata technology from Italy to Gaul through a multi-technique approach) the red color of this “terra sigillata” comes from the decomposition of the illite clay after firing and the formation of hematite with high degree of substitution by Al and Ti and high crystallinity. Since the presence of hematite is confirmed (Figure 1) Titanium analysis by ICP would have be interesting to perform on the samples.
Author’s reply: Thank you for your suggestion. The content of TiO2 of the clay minerals and the corresponding hybrid pigments can be determined by X-ray Fluorescence Spectrometry, and the results are shown in the Table R1 (was also supplemented in Table 1 in the revised manuscript). The TiO2 content in the hybrid pigments is small and it has no good correlation with the color value of the pigments. The content of titanium element in clay minerals is always higher than that in the corresponding hybrid pigments, indicating that titanium element is mainly in the crystalline framework of clay minerals, and is still present in the clay minerals after formation of hybrid pigments by a hydrothermal reaction process. The slight reduction of TiO2 content after hydrothermal reaction may be resulting from the dissolution of titanium element because that the metal cations may be removed by H+ ions in acid reaction condition. Perhaps, the effect of Titanium on the properties of the pigment can be demonstrated by doping the Titanium element during the synthesis of the hybrid pigment in our future work. Thanks for your helpful suggestion again, which would help us to improve our future work.
Table R1 TiO2 content of raw clay minerals and the corresponding hybrid pigments (mass%).
| TiO2 content (mass%) | |||||||
IS | REC | KAO | MMT | VMT | SEP | HYS | ILL | |
Clay minerals | 1.32 | 4.68 | 0.03 | 0.88 | 2.47 | 0.07 | -- | 1.32 |
Hybrid pigments | 0.84 | 3.84 | -- | 0.65 | 1.62 | -- | -- | 0.84 |
Question (2): Please try not to reproduce the content of the tables into the text of the discussion (lines 233-242). It is too repetitive.
Author’s reply: Thanks for your suggestion. We have removed the text that is duplicated in Table 1 according your suggestion.
In addition, the context was checked carefully and the improper phrase or sentences were corrected.
Finally, thanks for your intensive guidance to our manuscript. If you have any question about the revised manuscript, please do not hesitate to contact me.
Best Wishes
Sincerely
Prof. Aiqin Wang,
Lanzhou Institute of Chemical Physics,
Chinese Academy of Sciences,
Lanzhou 730000,
P. R. China,
Tel: 86 931 4968118
Fax: 86 931 4968019
E-mail: aqwang@licp.cas.cn

Round 2
Reviewer 2 Report
revised ms has been improved. it should be published